# Real-World Results of Stereotactic Body Radiotherapy for 399 Medically Operable Patients with Stage I Histology-Proven Non-Small Cell Lung Cancer

**DOI:** 10.3390/cancers15174382

**Published:** 2023-09-01

**Authors:** Hiroshi Onishi, Yoshiyuki Shioyama, Yasuo Matsumoto, Yukinori Matsuo, Akifumi Miyakawa, Hideomi Yamashita, Haruo Matsushita, Masahiko Aoki, Keiji Nihei, Tomoki Kimura, Hiromichi Ishiyama, Naoya Murakami, Kensei Nakata, Atsuya Takeda, Takashi Uno, Takuma Nomiya, Hiroshi Taguchi, Yuji Seo, Takafumi Komiyama, Kan Marino, Shinichi Aoki, Masaki Matsuda, Tomoko Akita, Masahide Saito

**Affiliations:** 1Department of Radiology, Faculty of Medicine, University of Yamanashi, 1110 Shimokato, Chuo-shi, Yamanashi 409-3898, Japan; takafumi@yamanashi.ac.jp (T.K.); marino@yamanashi.ac.jp (K.M.); aokis@yamanashi.ac.jp (S.A.); mmatsuda@yamanashi.ac.jp (M.M.); tsugiyama@yamanashi.ac.jp (T.A.); masahides@yamanashi.ac.jp (M.S.); 2Ion Beam Therapy Center, SAGA-HIMAT Foundation, 3049 Harakoga-machi, Tosu 841-0071, Japan; 3Department of Radiation Oncology, Niigata Cancer Center Hospital, 2-15-3 Kawagishi, Chuo-ku, Niigata 951-8566, Japan; ymatsu@niigata-cc.jp; 4Department of Radiation Oncology and Image-Applied Therapy, Graduate School of Medicine, Kyoto University, 54 Kawaharacho, Shogoin, Sakyo-ku, Kyoto 606-8507, Japan; ymatsuo@kuhp.kyoto-u.ac.jp; 5Department of Radiology, School of Medicine, Nagoya City University, 1 Kawasumi, Mizuho-cho, Mizuho-ku, Nagoya 467-8601, Japan; netakky115@hotmail.com; 6Department of Radiology, University of Tokyo, 7-3-1 Hongo, Bunkyo-ku, Tokyo 113-8655, Japan; yamachan07291973@yahoo.co.jp; 7Department of Radiation Oncology, School of Medicine, Tohoku University, 2-1 Seiryo-machi, Aoba-ku, Sendai 980-8574, Miyagi, Japan; harumatsu2001@yahoo.co.jp; 8Department of Radiation Oncology, Hirosaki University Graduate School of Medicine, 5 Zaifu-cho, Hirosaki City 036-8562, Aomori, Japan; maoki@hirosaki-u.ac.jp; 9Department of Radiation Oncology, Tokyo Metropolitan Cancer, Infectious Diseases Center Komagome Hospital, 3-18-22 Honkomagome, Bunkyo-ku, Tokyo 113-8677, Japan; knihei.cick@gmail.com; 10Department of Radiation Oncology, Hiroshima University, 1-2-3, Kasumi Minami-ku, Hiroshima 734-8551, Japan; tkkimura@kochi-u.ac.jp; 11Department of Radiation Oncology, Kitasato University School of Medicine, 1-15-1 Kitasato, Minami, Sagamihara 252-0375, Kanagawa, Japan; hishiyam@kitasato-u.ac.jp; 12Department of Radiation Oncology, National Cancer Center Hospital, 5-1-1 Tsukiji, Chuo-ku, Tokyo 104-0045, Japan; namuraka@ncc.go.jp; 13Department of Radiation Oncology, Sapporo Medical University, S1W17, Chuo-ku, Sapporo 060-8556, Hokkaido, Japan; kens72@yahoo.co.jp; 14Radiation Oncology Center, Ofuna Chuo Hospital, 6-2-24 Ofuna, Kamakura 247-0056, Kanagawa, Japan; takedaatsuya@gmail.com; 15Diagnostic Radiology and Radiation Oncology, Graduate School of Medicine, Chiba University, 1-8-1, Inohana, Chuo-ku, Chiba City 260-8670, Chiba, Japan; unotakas@faculty.chiba-u.jp; 16Department of Radiation Oncology, Yamagata University Faculty of Medicine, 2-2-2 Iida-Nishi, Yamagata-shi 990-9585, Yamagata, Japan; nomiya-rad@umin.ac.jp; 17Department of Radiation Oncology, Hokkaido University Hospital, North-14 West-5, Kita-ku, Sapporo 060-8648, Japan; t.hiroshi@pop.med.hokudai.ac.jp; 18Department of Radiation Oncology, Osaka University Graduate School of Medicine, 2-2 (D10) Yamada-oka, Suita 565-0871, Osaka, Japan; seo@radonc.med.osaka-u.ac.jp

**Keywords:** stereotactic body radiotherapy, stage I non-small cell lung cancer, operable, real-world evidence

## Abstract

**Simple Summary:**

According to current guidelines, surgery is the standard treatment for stage I non-small cell lung cancer (NSCLC), despite no clear randomized trial demonstrating that surgery is superior to stereotactic body radiotherapy (SBRT). Therefore, this study aimed to provide real-world evidence of the usefulness of SBRT for medically operable patients with pathologically proven stage I NSCLC using a large Japanese multi-institutional database. A total of 399 patients from 20 institutions were included in the database. In the results, local progression-free survival was 84.2%, and the 3-year overall survival was 77.0%. The local progression-free survival rate was better in cases with tumors ≤ 20 mm in diameter and in the adenocarcinoma subgroups. Low performance status, male sex, and pulmonary interstitial changes were poor prognostic factors for overall survival. This real-world evidence will be useful in shared decision-making for optimal treatment, including SBRT for operable stage I NSCLC, particularly in older patients.

**Abstract:**

Surgery is the standard treatment for stage I non-small cell lung cancer (NSCLC); however, no clear randomized trial demonstrates its superiority to stereotactic body radiotherapy (SBRT) regarding survival. We aimed to retrospectively evaluate the treatment outcomes of SBRT in operable patients with stage I NSCLC using a large Japanese multi-institutional database to show real-world outcome. Exactly 399 patients (median age 75 years; 262 males and 137 females) with stage I (IA 292, IB 107) histologically proven NSCLC (adenocarcinoma 267, squamous cell carcinoma 96, others 36) treated at 20 institutions were reviewed. SBRT was prescribed at a total dose of 48–70 Gy in 4–10 fractions. The median follow-up period was 38 months. Local progression-free survival rates were 84.2% in all patients and 86.1% in the T1, 78.6% in T2, 89.2% in adenocarcinoma, and 70.5% in squamous cell subgroups. Overall 3-year survival rates were 77.0% in all patients: 90.7% in females, 69.6% in males, and 41.2% in patients with pulmonary interstitial changes. Fatal radiation pneumonitis was observed in two patients, all of whom had pulmonary interstitial changes. This real-world evidence will be useful in shared decision-making for optimal treatment, including SBRT for operable stage I NSCLC, particularly in older patients.

## 1. Introduction

Lung cancer is the leading cause of death worldwide [1]. Early-stage lung cancer is increasingly detected using general computed tomography (CT) screening. Less invasive treatments are desirable for the current rapidly aging populations, especially in developed countries [2]. Stereotactic body radiotherapy (SBRT) has been actively used as a curative and less-invasive treatment option for patients with stage I NSCLC [3,4,5,6,7]. As the standard treatment for operable early-stage NSCLC is surgery [8], most patients treated with SBRT are medically inoperable or high-risk. However, some medically operable patients do not favor surgery for personal reasons. The use of SBRT has increased, mostly among older patients with smaller tumors or a high comorbidity index and the increase in SBRT has contributed to the significant decrease in patients who had no therapy in the US [9]. A nationwide population-based study from the Netherlands showed that the application rates of surgery and radiation therapy for stage I NSCLC have been reversed and the most popular treatment has recently been established as SBRT [10]. The results of a pooled analysis of randomized trials comparing SBRT versus surgery (STARS and ROSEL) suggested that SBRT is a treatment option for operable patients requiring a lobectomy [11]. However, the evidence level is considered poor due to the study design, the pooled analysis of two incomplete trials, and the small sample size. Additional prospective randomized clinical trials comparing SBRT with surgery are in progress, but more time is needed to determine their results.

Randomized controlled trials (RCTs) have long been considered the gold standard for evaluating medical interventions. However, RCTs often come with limitations, including strict inclusion criteria, short follow-up periods, and potential lack of representation of real-world patient populations. Recently, there has been an increased interest in understanding the potential benefits of generating real-world evidence [12,13,14], particularly when it is difficult to complete randomized trials. Real-world evidence addresses these shortcomings by providing data and information collected from routine clinical practice, electronic health records, observational studies, and patient registries. As SBRT has been developed and actively practiced widely in Japan, we organized a multi-institutional SBRT study group at the Japanese Radiological Society (JRS-SBRTSG) to conduct a retrospective study. This study aimed to investigate the real-world results of SBRT in patients with operable stage I NSCLC using a large database. There are two articles in the field of SBRT for lung cancer containing real-world evidence [15,16], however, these studies did not include the information on operability. Therefore, this is the first report to refer to real-world evidence of SBRT for operable stage I NSCLC.

## 2. Materials and Methods

### 2.1. Study Design

We organized a multi-institutional SBRT study group at the Japanese Radiological Society (JRS-SBRTSG) and conducted a retrospective, multi-institutional study. The original JRS-SBRTSG database included data from over 2000 patients from 20 participating institutions treated with SBRT for stage I NSCLC before 2017. The participating institutions from which data were collected were selected because they actively performed SBRT in Japan and demonstrated a willingness to cooperate and register more than 20 cases in this study. The data were centralized and recorded electronically at the principal research institution. All patients who satisfied the following eligibility criteria were included: (1) identification of T1N0M0 or T2N0M0 primary lung cancer based on the criteria of the Union for International Cancer Control version 7 on chest and abdominal CT, positron emission tomography, bone scintigraphy, and/or brain magnetic resonance imaging; (2) histopathological confirmation of NSCLC; and (3) medically operable status but refusal to undergo surgery and the selection of SBRT. Medical operability was determined by the multidisciplinary tumor board of each institution based on respiratory function, age, and complications. To be deemed “medically operable,” patients had to meet all of the following criteria: World Health Organization performance status of 2 or less, arterial oxygen pressure > 75 mm Hg, a predicted postoperative forced expiratory volume in 1 s of >800 mL, and the absence of severe heart failure, diabetes mellitus, or arrhythmias. The decision to refuse surgery was made by the patient after consultation with the pulmonary surgeon or pulmonologist. This process was recorded in the chart.

### 2.2. Patients Characteristics

In total, 399 medically operable patients were enrolled in this study. A summary of patients’ pretreatment characteristics is presented in Table 1.

### 2.3. Treatment Methods

SBRT techniques differed among institutions. The technical details of SBRT at the participating 20 institutions are shown in Table 2, including clinical target volume (CTV) margin, immobilization method, breathing motion management, planning target volume-CTV margin, image guidance method, irradiating ports, calculation algorithm, prescription point, and the irradiated dose/fractionations. The gross target volume was delineated on CT images using a lung window level setting. A total dose of 48–70 Gy in 4–10 fractions was delivered mainly at the isocenter within 4–25 days, with 6 MV X-rays. The median calculated biologically effective dose was 105.6 Gy (range, 100.0–150.0 Gy) based on alpha/beta = 10 Gy.

### 2.4. Evaluation and STATISTICS 

The follow-up period began at the beginning of SBRT. Local control (LC), the same as local progression-free survival, was defined as the irradiated tumor without enlargement of the local tumor that continued for 6 months as determined by follow-up CT or histological confirmation. Overall survival (OS), LC, and prognostic factors for OS were analyzed. Examined prognostic factors were sex, age (≥75 years versus <75 years), performance status (0 versus 1–2), pathology (adenocarcinoma versus squamous cell carcinoma), stage (IA versus IB), pulmonary emphysema, and pulmonary interstitial change on CT. CT findings on the lungs were obtained by a diagnostic radiologist. The survival rate was calculated using the Kaplan–Meier method. Statistical differences were calculated using the log-rank test in the univariate analysis and the Cox proportional hazards model in the multivariate analysis. Differences were considered statistically significant at *p*-values < 0.05. All statistical analyses were performed using StatView software version 5.0 (SAS, Cary, NC, USA). Toxicity occurring during or after SBRT was graded according to the Common Terminology Criteria for Adverse Events (CTCAE), version 3.0.

## 3. Results

The median follow-up period for all patients was 38 months (range, 1–144 months). Overall, 98% of patients were observed for more than 6 months.

### 3.1. Local Control and Survival

Figure 1 shows the LC and OS curves of all cases, respectively. The 3- and 5-year LC rates were 84.2 (95% confidence interval [CI], 80.0–88.4)% and 79.9 (CI, 74.5–85.4)%, respectively. The 3- and 5-year OS rates were 77.7 (95% CI, 72.4–81.6)% and 64.3 (95% CI, 58.1–70.5)%, respectively. Figure 2 shows the LC rates according to subgroups arranged by tumor size (maximum diameter less than or equal to 20 mm versus more than 20 mm) and histology (adenocarcinoma versus squamous cell carcinoma). Figure 3 shows the OS according to sex (male versus female), age (less than 75 years versus ≥75 years), tumor size (maximum diameter ≤ 20 mm versus > 20 mm), histology (adenocarcinoma versus squamous cell carcinoma), performance status (0 versus 1, 2, 3), existence of pulmonary emphysema (negative versus positive), and existence of pulmonary interstitial changes (negative versus positive).

The results of the univariate and multivariate analyses are presented in Table 3. Univariate analysis revealed a significantly better OS in patients with performance status 0, female sex, adenocarcinoma, no pulmonary emphysema, and no pulmonary interstitial changes. Multivariate analysis revealed significantly better OS in the subgroups with performance status 0, female sex, and no pulmonary interstitial changes. 

### 3.2. Toxicity

Radiation-induced pneumonitis of >Grade 3 according to the CTCAE version 3.0 was observed in 10 (2.5%) patients; 2 (0.5%) were Grade 5. The two patients with Grade 5 radiation-induced pneumonitis had pulmonary interstitial change on CT before SBRT. The frequency of other toxicities was unclear due to the retrospective nature of this study. However, no other grade 3 or higher toxicities were reported. 

## 4. Discussion

To the best of our knowledge, this is the first report to refer to real-world evidence on SBRT for operable stage I NSCLS. Table 4 summarizes the results of SBRT for operable early-stage lung cancer. Onishi et al. reported the 5-year OS of 87 operable patients as 72.0% (95% confidence interval; CI 59.6–84.4%) after SBRT with a biologically effective dose > 100 Gy in a multi-institutional retrospective study [16]. Nagata et al. first reported the results of a phase II trial on SBRT in patients with stage IA NSCLC in 2013. In the study, the 3-year OS of 64 operable patients was 76.5% (95% confidence interval; CI 64.0%–85.1%) after SBRT of 48 Gy in four fractions [17]. Timmerman et al. also reported that the 4-year OS of 26 operable patients was 56% (95% CI, 35%–73%) after a phase II trial administering SBRT of 54 Gy in three fractions [18]. These OS data were inferior to those of general surgical reports. However, the SBRT series had disadvantages regarding age and performance status.

Tandberg et al. conducted a comprehensive review comparing SBRT and surgery for stage I NSCLC [19] and concluded that OS was better with surgery, but cause-specific survival outcome was comparable. They speculated that differences in patient selection due to age and comorbidities between surgical and SBRT cohorts were likely strong contributors to the result. 

Many propensity-matched analyses compared SBRT and surgery, but the results varied [20,21,22,23,24]. Chen et al. conducted a meta-analysis of propensity score studies [25]. They concluded that better OS was seen after surgery compared with Stereotactic Ablative Radiotherapy (SABR), but lung cancer-specific survival was similar for both treatments. They found that the survival results differed according to the specialty of the first author.

**Table 4 cancers-15-04382-t004:** Results of stereotactic body radiotherapy for operable patients with stage I non-small cell lung cancer.

Author [References No.]	Type of Study	Pt. No (Stage)	Age Min–Max (Median)	Dose Gy/ Fraction (fx)	Median Follow-Up (Months)	Local Control	Overall Survival
Onishi H. [16]	Retrospective	87 (IA 65, IB 22)	51–87 (77)	45–72.5 Gy/3–10 fr	55	IA 92%, IB 73% (at 5-year)	IA 72%, IB 62% (at 5-year)
Nagata Y. [17]	Prospective	IA 64	50–91 (79)	48 Gy/4 fr	67	85.4% (at 3-year)	76.5% (at 3-year)
Timmerman R. [18]	Prospective	26 (IA 23, IB 3)	54–88 (72)	54 Gy/3 fr	48	96% (at 4-year)	56% (at 4-year)
Chang J. [11]	Prospective	31 (IA 27, IB 4)	43–82 (67)	54 Gy/3 fr (peripheral) 50 Gy/4 fr (central)	40	96% (at 3-year)	95% (at 3-year)
Chang J. [26]	Prospective	IA 80	Mean 69	54 Gy/3 fr (peripheral) 50–60 Gy/4 fr (central)	61	not shown	87% (at 5-year)
This study	Retrospective	399 (IA 292, IB 107)	40–89 (75)	48–70 Gy/4–10 fr	38	IA 86.2%, IB 78.4% (at 3-year)	IA 78.6%, IB 75.1% (at 3-year)

Although some randomized trials comparing SBRT and surgery were conducted, most were terminated prematurely because of poor patient recruitment. Chang et al. combined the results of two randomized phase III trials in patients with operable stage I NSCLC (STARS and ROSEL) and estimated that OS at 3 years was 95% (95% CI 85–100) in the SBRT group compared with 79% (95% CI 64–97) in the surgery (lobectomy and mediastinal) group (hazard ratio [HR] 0·14 [95% CI 0·017–1.190], log-rank *p* = 0·037) [11]. However, it did not lead to changes in the standard treatment guidelines because of the low level of evidence. Chang et al. also reported long-term results of a single-arm STARS prospective trial that showed 87% 5-year OS in 80 patients after SBRT of 54 Gy in three fractions (for peripheral lesions) or 50 Gy in four fractions (for central tumors; simultaneous integrated boost to gross tumor totaling 60 Gy) and concluded that SBRT was non-inferior to surgery [26]. Cao et al. conducted a systematic review and meta-analysis and concluded that surgery was superior to SBRT in terms of mid- and long-term clinical outcomes and SBRT is associated with lower perioperative mortality [27]. However, they considered that this may be attributed, at least in part, to an imbalance in baseline characteristics, and SBRT was associated with lower perioperative mortality. There are four ongoing randomized studies comparing SBRT with surgery for early-stage lung cancer: NCT02984761; Veterans Affairs Lung Cancer Surgery Or Stereotactic Radiotherapy (VALOR), NCT02984761; JoLT-Ca Sublobar Resection versus Stereotactic Ablative Radiotherapy (SABR) for Lung Cancer (STABLE-MATES), NCT02468024; Radical Resection vs. Ablative Stereotactic Radiotherapy in Patients with Operable Stage I NSCLC (POSTILV), and NCT01753414; and SABR versus surgery for stage I NSCLC (SABRTooth). The findings of these studies have not been presented. 

Recently, a better prognosis after segmentectomy over lobectomy for peripheral small NSCLC by a randomized study was reported [28,29]. The results mean that the focal treatment is acceptable and will assist the rationale of future large trials comparing surgery and SBRT for peripheral small NSCLC. 

OS is generally worse in older patients owing to other causes of death. The median age in this study was 75 years, which was higher than the average age in general surgical series. Okami et al. [30] and Matsuoka et al. [31] reported the 3-year OS rates of octogenarians (more than 80 years old) as 70.6% and 71.6%, respectively. As there was no significant difference in OS between the two age subgroups (less than 75 years versus 75 years), the 3-year OS in this study could be considered comparable to that with surgery. 

According to the endorsement by the American Society of Clinical Oncology of the American Society for Radiation Oncology evidence-based guidelines, patients with stage I NSCLC should be evaluated by a thoracic surgeon, preferably within a multidisciplinary cancer care team to determine operability, and SBRT is not recommended as an alternative to surgery outside of a clinical trial for patients with standard operative risk and stage I NSCLC. Patients should be informed that, while SBRT may have decreased risks in the short term, long-term outcomes (>3 years) are not well established. This endorsement has not been revised since 2017, regardless of new or updated information [32].

Our study findings demonstrate that the OS rate of SBRT for medically operable and older patients is not inferior to that of surgery in the real world, even though SBRT might have some disadvantages in terms of patient and tumor characteristics, such as performance status, comorbidities, or grade of cancer cell malignancy. These analyses do not provide sufficient data to change the standard of care for good surgical candidates, but do help to confirm the indication for SBRT in patients with relative contraindications to surgery or those who refuse surgery. There is no evidence that SBRT is equivalent to lobectomy for patients with operable early-stage NSCLC, but there is also no evidence that SBRT is inferior to lobectomy. Therefore, the decision-making process should be shared with the patient. Pulmonologists must explain SBRT as an optional treatment for patients regardless of operability, and detailed information on SBRT should be provided by radiation oncologists. Then, the shared decision-making process is considered more important, particularly in elderly patients, because they tend to hope for less-invasive treatment such as SBRT and because the age-related mortality associated with pulmonary surgery increases with age [33,34]. 

Concerning prognostic factors affecting survival, male and pulmonary interstitial change were associated with poor outcomes in this study. Pulmonary interstitial change was reported as a high-risk factor for fatal radiation pneumonitis after SBRT [35]. Thus, it is important to consider this finding, even in operable patients. 

This study had some limitations. First, this was a retrospective study without unified criteria or a central review for judging medical operability and lacked the quality assurance of SBRT. This is a major problem of the study and therefore it could not propose an ideal SBRT method for clinical studies comparing SBRT and surgery. Second, the study had some missing data; however, the information on survival was confirmed for all enrolled patients, implying that the OS values were considered reliable. Third, the SBRT method differed among institutions; however, this is why the results are meaningful as real-world evidence. Finally, a larger number of patients and longer follow-up duration are mandatory to obtain persuasive real-world evidence. 

## 5. Conclusions

The survival rates of operable patients with stage I NSCLC treated with SBRT extracted from a real-world multi-institutional database were good and might not be inferior to those obtained by surgical intervention in older patients. This study provides useful information supporting a shared decision-making process for patients who are confused about whether to undergo surgery.

## Figures and Tables

**Figure 1 cancers-15-04382-f001:**
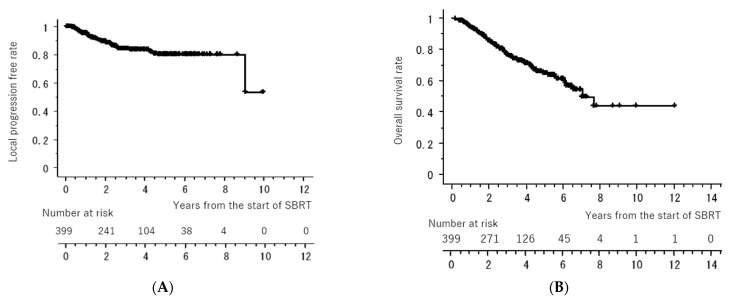
Local control rate (**A**) and overall survival rate (**B**) of all cases. All of the “+” on the curve mean censored cases.

**Figure 2 cancers-15-04382-f002:**
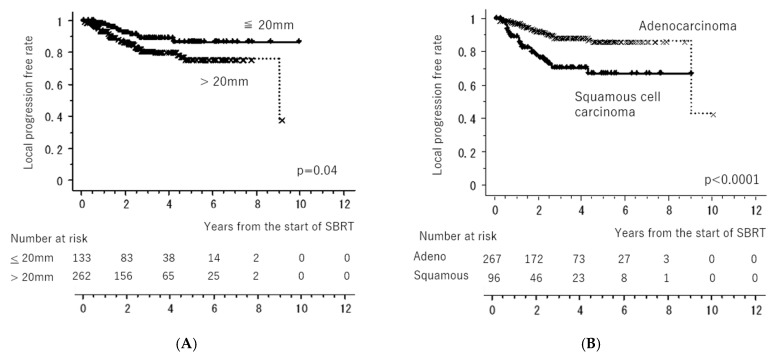
Local control rates according to subgroups arranged by tumor size ((**A**): maximum diameter less than or equal to 20 mm versus more than 20 mm) (**A**) and histology ((**B**): adenocarcinoma versus squamous cell carcinoma). All of the “x” and “+” on the curves mean censored cases.

**Figure 3 cancers-15-04382-f003:**
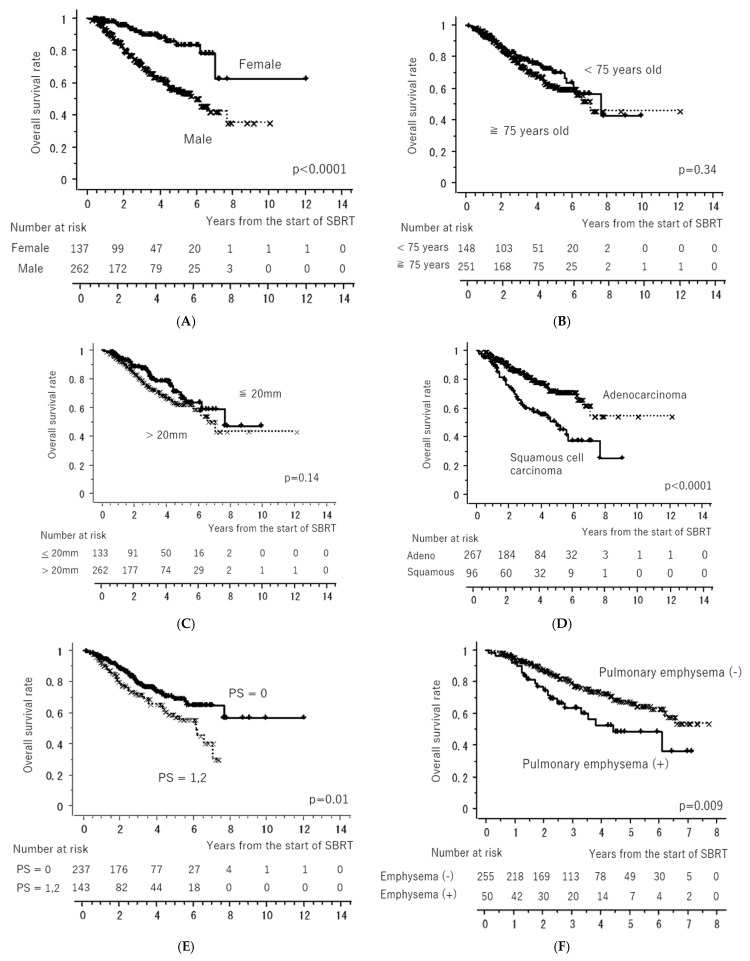
Overall survival rates according to sex ((**A**): male versus female), age ((**B**): less than 75 years old versus more than or equal to 75 years old), tumor size ((**C**): maximum diameter less than or equal to 20 mm versus more than 20 mm), histology ((**D**): adenocarcinoma versus squamous cell carcinoma), performance status ((**E**): 0 versus 1, 2), existence of pulmonary emphysema ((**F**): negative versus positive), and existence of pulmonary interstitial changes ((**G**): negative versus positive). All of the “x” and “+” on the curves mean censored cases.

**Table 1 cancers-15-04382-t001:** Characteristic of the patients and radiotherapy in 20 institutions.

Characteristics	Total (*n* = 399)
Age, years	Median 75 (range: 40–89)
Sex	
Male	262
Female	137
Performance status	
ECOG score 0	237
ECOG score 1	129
ECOG score 2	14
ECOG score unknown	19
Pathology	
Adenocarcinoma	267
Squamous cell carcinoma	96
Other type of non-small cell lung cancer	36
Disease stage (UICC Version 7)	
Stage IA	292
Stage IB	107
Tumor diameter, mm	Mean 26 (range: 5–55)
Pulmonary emphysema	
(−)	255
(+)	50
Unknown	94
Pulmonary interstitial change	
(−)	292
(+)	31
Unknown	76

**Table 2 cancers-15-04382-t002:** Technical details of SBRT at 20 institutions.

SBRT Technique	Number of Institutions (*n* = 20)
CTV margin (mm)	
0	14
1–3	2
5	3
6–8	1
Immobilization	
Vacuum-type pillow	14
Body-frame	4
Shell-type	2
Breathing motion management	
Breath-hold	8
Abdominal compression	6
Gating	2
Real-time tumor tracking	1
Free	3
PTV–CTV margin (mm)	
5 (all directions)	18
Cranio-caudal 6, others 5	1
Cranio-caudal 8, dorsal 7, others 6	1
Image guidance method	
On board cone-beam CT	9
Portal image	6
In-room CT	2
ExacTrac	2
Real-time tumor tracking	1
Ports	
Multiple static ports	18
Rotational arcs	2
Energy of X-ray	
6 MV	17
6, 10 MV	3
Calculation algorithm	
Superposition	10
AAA	6
Monte Carlo	2
CC convolution	2
Prescription point	
Isocenter (80–90% isodose covered PTV)	16
PTV periphery	4
Dose (Gy)/fractionations	
48–52/4	17
60–70/10	2
Others	1

**Table 3 cancers-15-04382-t003:** Results of the univariate and multivariate analyses for overall survival.

Variables		Univariate Analysis	Multivariate Analysis	
			Hazard Ratio (95% CI *)	*p*-Value
PS	0 vs. 1, 2	0.01	0.62 (0.38–1.01)	0.05
Gender	Female vs. Male	<0.01	0.25 (0.12–0.53)	<0.01
Age	<75 years vs. ≥75 years	0.42	0.65 (0.38–1.12)	0.11
Stage	IA vs. IB	0.14	1.02 (0.60–1.75)	0.93
Histology	Squamous cell carcinoma vs. Adenocarcinoma	<0.01	1.25 (0.74–2.10)	0.40
Pulmonary emphysema	(+) vs. (−)	<0.01	0.75 (0.39–1.43)	0.38
Pulmonary interstitial change	(+) vs. (−)	<0.01	4.88 (2.47–9.61)	<0.01

* CI: confidence interval

## Data Availability

The data presented in this study are available upon reasonable request from the corresponding author.

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
