# Peer review of "Real-World Results of Stereotactic Body Radiotherapy for 399 Medically Operable Patients with Stage I Histology-Proven Non-Small Cell Lung Cancer"

_cancers, 2023, doi:10.3390/cancers15174382_

Round 1

Reviewer 1 Report

Thank you for submitting this report.

The number of cases collected over the years is good, and I think these data can be useful in integrating the current knowledge on SBRT outcome for operable patients,

Please better describe/clarify:

1. methods: how did you collect all the data? centralized electronic records? how were Centres selected? was there a minimum number of cases for each center? it is "real-world", but Centres could have been selected according to many factors, please clarify

2. was the definition of "medically operable but refusing surgery" clear and the same for all? could you please clarify what the definition was and how you were confident that every Centre did enroll patients who actually refused surgery?

3. technical issues: as the number of centers I high and heterogenous, how could you say, for example, that the margin for PTV was 5 mm? in all 20 Centers? it seems quite unlikely. Please report in a table all the technical details, with all ranges (margins, dose prescription, fractionation, technique)

4. please discuss the results in light of the new surgical standard of segmentectomy and better describe the current ongoing trials comparing SBRT to surgery (for example, the VALOR study). In all these trials, quality assurance for SBRT and surgery is mandatory. In your study, please highlight that one of the main limitations is that you can't have any control over the quality of SBRT, and limited control on patients' selection.

5. the follow-up range is very wide, from 1 to 144 months. It is difficult to extrapolate data for patients who have been followed for very few months; however, the median is 38: when does the observation begin? from diagnosis or from the end of SBRT?

It can be improved, but acceptable.

Author Response

Date of submission: 9th August, 2023

Dear Reviewer 1,

I appreciate your valuable comments so much, I have revised my manuscript according to your comments and wish to submit the new manuscript for an original article for publication in Cancers, titled “Real-world results of stereotactic body radiotherapy for 399 medically operable patients with stage I histology-proven non-small cell lung cancer.” The paper was coauthored by Yoshiyuki Shioyama, Yasuo Matsumoto, Yukinori Matsuo, Akifumi Miyakawa, Hideomi Yamashita, Haruo Matsushita, Masahiko Aoki, Keiji Nihei, Tomoki Kimura, Hiromichi Ishiyama, Naoya Murakami, Kensei Nakata, Atsuya Takeda, Takashi Uno, Takuma Nomiya, Tsuyoshi Takanaka, Hiroshi Taguchi, Yuji Seo Takafumi Komiyama, Kan Marino, Shinichi Aoki, Masaki Matsuda, Tomoko Akita, and Masahide Saito.

We are grateful for the opportunity to submit our revised manuscript. Your suggestions were valuable and we have taken them into account. We believe that our manuscript has been strengthened immensely from your insightful suggestions and careful reviews.

Please see the revised manuscript in the attached file with our point-by-point responses to your comments.

I hope that the revised manuscript is now suitable for publication.

This manuscript has not been published or presented elsewhere in part or in entirety and is not under consideration by another journal. All study participants provided informed consent, and the study design was approved by the appropriate ethics review board. We have read and understood your journal’s policies, and we believe that neither the manuscript nor the study violates any of these. There are no conflicts of interest to declare.

Thank you for your consideration. I look forward to hearing from you.

Sincerely,

Hiroshi Onishi,

Department of Radiology, School of Medicine, University of Yamanashi,

1110 Shimokato, Chuo City, Yamanashi  409-3898, Japan

Tel: 81-55-273-1111, ext 2382; Fax: 81-55-273-9766 

Reviewer‘s comments and the point-by-point author’s responses:

Reviewer 1

The number of cases collected over the years is good, and I think these data can be useful in integrating the current knowledge on SBRT outcome for operable patients, Please better describe/clarify:

Comment 1: Methods: how did you collect all the data? centralized electronic records? how were Centres selected? was there a minimum number of cases for each center? it is "real-world", but Centres could have been selected according to many factors, please clarify.

Response: We appreciate this valuable comment. To address this, we have included the following sentences describing the process of data collection under the Methods: "The participating institutions from which data were collected were selected because they actively performed SBRT in Japan and demonstrated a willingness to cooperate and register more than 20 cases in this study. The data was centralized and recorded electronically in the principal research institution.

Comment 2: Was the definition of "medically operable but refusing surgery" clear and the same for all? could you please clarify what the definition was and how you were confident that every Centre did enroll patients who actually refused surgery?

Response: We appreciate this valuable comment. To address this, we have included the following sentences in the Methods describing the verification of refusal to undergo surgery: "The decision to refuse surgery was made by the patient after a consultation with the pulmonary surgeon or pulmonologist. This process was recorded in the chart.

Comment 3: Technical issues: as the number of centers I high and heterogenous, how could you say, for example, that the margin for PTV was 5 mm? in all 20 Centers? it seems quite unlikely. Please report in a table all the technical details, with all ranges (margins, dose prescription, fractionation, technique)

Response: We appreciate this valuable comment. Accordingly, we have included a table showing the technical details of the SBRT method (Table 2).

Comment 4: Please discuss the results in light of the new surgical standard of segmentectomy and better describe the current ongoing trials comparing SBRT to surgery (for example, the VALOR study). In all these trials, quality assurance for SBRT and surgery is mandatory. In your study, please highlight that one of the main limitations is that you can't have any control over the quality of SBRT, and limited control on patients' selection.

Response: We appreciate this valuable comment. To address this, we have included additional sentences describing the ongoing trials comparing SBRT to surgery. We also referred to the recent trial showing a better prognosis with segmentectomy over lobectomy for peripheral small NSCLC, and the result will assist the rationale of the future large trial comparing surgery and SBRT for peripheral small NSCLC. Finally, we included this sentence under the study limitations: "First, this was a retrospective study without unified criteria or a central review for judging medical operability and lacked quality assurance of SBRT. This is a main problem of the study and it could not propose an ideal SBRT method for clinical studies comparing SBRT and surgery.

Comment 5: The follow-up range is very wide, from 1 to 144 months. It is difficult to extrapolate data for patients who have been followed for very few months; however, the median is 38: when does the observation begin? from diagnosis or from the end of SBRT?

Response: We appreciate this valuable comment. We have now included that the follow-up period began at the commencement of SBRT (page 5, line 139). Some patients died of other diseases or complications within a short time after SBRT, possibly due to high age; therefore, we included all patients regardless of the length of follow-up duration. The reason for the very short follow-up duration in some cases was death from other diseases. We have also included that 98% of the patients were observed for more than 6 months.

Other revisions: We have added some sentences to expand the introduction and discussion sections in order to satisfy the manuscript requirement. The numbers of the references have been modified according to the revision.

Reviewer 2 Report

Conceptually interesting work, with large real word case series from a multi-institutional study group, however retrospective and therefore not homogeneous in terms of baseline selection criteria and treatment technique.

The article lacks salient technical data on radiation treatment: the relative iconography is also lacking.
The bibliography is sufficient.
The figures and tables are exhaustive and well described in the text and captions.

In the manuscript I notice the following problems:
- number of patients: 419 in “Patient Characteristics” paragraph, vs 399 in Abstract and in Table 1: why?
- CTCAE scale: version 4 in “Evaluation and Statistics”  paragraph, vs version 3 (named National Cancer Institute-Common Toxicity Criteria) in “Toxicity” paragraph: which one was used?
- bibliographic citation n°3 incorrect, both in the text (Discussion paragraph) and in Table 3.
- some errors to be corrected:
o  line 198: add "multivariate"
o  line 242: delete [10] and period
o  line 271: delete "a"
o  line 283: delete "s” from the noun “durations”

My suggestion is to search for the missing technical data and to insert them in the work, both in the text and also in table form, describing the main parameters used in the treated population such as:
volumes, doses (total, fraction, equivalent), beam energy, immobilization and target motion control system, simulation and treatment technique…

The article therefore can be evaluated for publication after major revision.

In the manuscript there are some spelling and grammar errors that need to be checked and corrected.

Author Response

Date of submission: 9th August, 2023

Dear Reviewer 2,

I appreciate your valuable comments so much, I have revised my manuscript according to your comments and wish to submit the new manuscript for an original article for publication in Cancers, titled “Real-world results of stereotactic body radiotherapy for 399 medically operable patients with stage I histology-proven non-small cell lung cancer.” The paper was coauthored by Yoshiyuki Shioyama, Yasuo Matsumoto, Yukinori Matsuo, Akifumi Miyakawa, Hideomi Yamashita, Haruo Matsushita, Masahiko Aoki, Keiji Nihei, Tomoki Kimura, Hiromichi Ishiyama, Naoya Murakami, Kensei Nakata, Atsuya Takeda, Takashi Uno, Takuma Nomiya, Tsuyoshi Takanaka, Hiroshi Taguchi, Yuji Seo Takafumi Komiyama, Kan Marino, Shinichi Aoki, Masaki Matsuda, Tomoko Akita, and Masahide Saito.

We are grateful for the opportunity to submit our revised manuscript. Your suggestions were valuable and we have taken them into account. We believe that our manuscript has been strengthened immensely from your insightful suggestions and careful reviews.

Please see the revised manuscript in the attached file with our point-by-point responses to your comments.

I hope that the revised manuscript is now suitable for publication.

This manuscript has not been published or presented elsewhere in part or in entirety and is not under consideration by another journal. All study participants provided informed consent, and the study design was approved by the appropriate ethics review board. We have read and understood your journal’s policies, and we believe that neither the manuscript nor the study violates any of these. There are no conflicts of interest to declare.

Thank you for your consideration. I look forward to hearing from you.

Sincerely,

Hiroshi Onishi,

Department of Radiology, School of Medicine, University of Yamanashi,

1110 Shimokato, Chuo City, Yamanashi  409-3898, Japan

Tel: 81-55-273-1111, ext 2382; Fax: 81-55-273-9766 

Reviewer‘s comments and the point-by-point author’s responses:

Reviewer2

Conceptually interesting work, with large real word case series from a multi-institutional study group, however retrospective and therefore not homogeneous in terms of baseline selection criteria and treatment technique.

Comment: The article lacks salient technical data on radiation treatment: the relative iconography is also lacking.

The bibliography is sufficient.

The figures and tables are exhaustive and well described in the text and captions.

Comment: In the manuscript I notice the following problems: - number of patients: 419 in "Patient Characteristics" paragraph, vs 399 in Abstract and in Table 1: why?

Response: We appreciate this valuable comment. The inclusion of 419 was an error and has been revised to 399.

Comment: CTCAE scale: version 4 in "Evaluation and Statistics" paragraph, vs version 3 (named National Cancer Institute-Common Toxicity Criteria) in "Toxicity" paragraph: which one was used?

Response: Thank you for this valuable comment. CTCAE version 3.0 was used. This has been revised throughout the manuscript.

Comment: Bibliographic citation n°3 incorrect, both in the text (Discussion paragraph) and in Table 3.

Response: We appreciate this valuable comment. However, the reference number in the introduction, discussion, and Table 3 is the same, and reference "3" is correct: (3. Chang J.Y.; Senan S.; Paul M.A.; Mehran R.J.; Louie A.V.; Balter F.; Groen H.J.H.; McRae S.E.; Widder J.; Feng L.; Borne  E.E.E.M.; et al. Stereotactic ablative radiotherapy versus lobectomy for operable stage I non-small-cell lung cancer: a pooled analysis of two randomised trials. Lancet Oncol, 2015, 16, 630-7.) (The number of the reference has been changed from 3 to 11. The number of the table has been changed from 3 to 4.)

Comments: some errors to be corrected:

o  line 198: add "multivariate"

o  line 242: delete [10] and period

o  line 271: delete "a"

o  line 283: delete "s" from the noun "durations"

Response: Thank you very much for the corrections. We appreciate this feedback. These errors have been corrected accordingly.

Comment: My suggestion is to search for the missing technical data and to insert them in the work, both in the text and also in table form, describing the main parameters used in the treated population such as: volumes, doses (total, fraction, equivalent), beam energy, immobilization and target motion control system, simulation and treatment technique…

Response: Thank you very much for this valuable comment. To address this, we included a table showing the technical details of the SBRT method, including these parameters (Table 2).

The article therefore can be evaluated for publication after major revision.

Comments on the Quality of English Language

In the manuscript there are some spelling and grammar errors that need to be checked and corrected.

Response: I appreciate this valuable comment. To address this, we have ensured that the revised manuscript was edited by a native English speaker.

Other revisions: We have added some sentences to expand the introduction and discussion sections in order to satisfy the manuscript requirement. The numbers of the references have been modified according to the revision.

Round 2

Reviewer 2 Report

I thank the Authors for their work, in agreement with the comments made in the first revision of the manuscript.

As far as I am concerned, I believe that all requests have been processed: therefore, in my opinion, the article can now be published.